# Torsional Fatigue Life Prediction of 30CrMnSiNi2A Based on Meso-Inhomogeneous Deformation

**DOI:** 10.3390/ma14081846

**Published:** 2021-04-08

**Authors:** Cheng-Xian Cen, Da-Min Lu, Da-Wei Qin, Ke-Shi Zhang

**Affiliations:** 1Key Laboratory of Disaster Prevention and Structural Safety/Guangxi Key Lab Disaster Prevention and Engineering Safety, College of Civil and Architectural Engineering, Guangxi University, Nanning 530004, China; cenchengxian@st.gxu.edu.cn (C.-X.C.); ludamin@st.gxu.edu.cn (D.-M.L.); 2Guangdong Provincial Key Laboratory of Durability for Marine Civil Engineering, College of Civil and Transportation Engineering, Shenzhen University, Shenzhen 518060, China; qindawei@szu.edu.cn

**Keywords:** torsional fatigue, non-masing behavior, inhomogeneous deformation, crystal plasticity, fatigue indicator parameter (FIP), fatigue life prediction

## Abstract

In this paper, torsional fatigue failure of 30CrMnSiNi2A steel which exhibited non-Masing behavior was studied under different constant shear strain amplitudes, using thin-walled tubular specimens. The relationship between shear fatigue and the evolution of meso-deformation inhomogeneity and the prediction method of the torsional fatigue life curve were investigated. Shear fatigue of the material under constant amplitude was researched by numerical simulation with reference to tests, by using crystal plasticity of polycrystalline representative volume element (RVE) as the material model. Considering the non-Masing behavior of material, when determining the parameter values of the crystal plasticity model the correlation between these parameters and strain amplitude was taken into account. The meso-deformation inhomogeneity with increments in the number of cycles was characterized by using the statistical shear strain standard deviation of RVE as the basic parameter. Considering the effect of strain amplitude on fatigue damage, ratio cycle peak stress/yield stress was taken as the weight to measure the torsional fatigue damage and an improved fatigue indicator parameter (FIP) to measure the inhomogeneous deformation of the material was proposed. The torsional fatigue life curve of 30CrMnSiNi2A steel was predicted by the critical value of the FIP and then the result was confirmed.

## 1. Introduction

Shaft components, such as machine drive shafts, are subjected to torsional loading that results in fatigue failure. The hysteresis behavior of the torsional cycle is comparable to that of the tension–compression cycle [1,2,3,4]. Furthermore, some materials exhibit non-Masing behavior under torsional cyclic loading [5]. The investigations reported in references [2,6,7,8] found that the hysteresis loop of the material under axial load and shear load is different, and the mechanism of crack initiation and propagation is also different. For pure torsional fatigue, fatigue cracks usually initiate on the surface or near the surface of the specimen, propagating parallel or perpendicular to the torsion axis of the specimen [3,6,7,8,9,10,11]. For solid specimens, some researchers observe that the torsional fatigue global fracture surface present concentric ring-like features, its center is the final rupture region and, moreover, the fatigue crack source is distributed on the specimen’s surface [12,13].

For torsional fatigue tests, the relationship between cyclic shear stress and shear strain of materials is represented by the Ramberg–Osgood equation commonly [3,14], but the relationship between shear strain amplitude and fatigue life is described by the Manson–Coffin or Morrow equation [11,14]. In order to predict the torsional fatigue life of the material, it is always necessary to obtain the torsional fatigue life curve which must first be fitting through a series of test data [15,16]. With the development of computational technology, the numerical simulation method has been widely used in the study of the fatigue failure mechanism in recent years. Some scholars propose some fatigue life analysis models which used cumulative plastic slip and energy dissipation calculated by crystal plasticity [17], plastic work [18,19], equivalent plastic strain [20] as parameters, etc. For the polycrystalline metals, many researchers suggest the material model of representative volume element (RVE) constructed by using the Voronoi polycrystalline polyhedron aggregation [21,22] and study the mesoscopic behavior of materials under cyclic loading with crystal plasticity.

The stress–strain hysteresis behavior observed in tests can be simulated by modelling with classical continuum mechanics. However, the results can reflect the apparent stress and strain responses of homogenized materials. Given the complexity and irregularity of actual metal material structure and the anisotropy of mechanical properties, the deformation and stress distribution of the materials are non-uniform and their evolution is dependent on the change of the materials’ microstructure. Hence, the mechanics analysis should consider of the materials’ structure. In order to investigate the correlation between deformation inhomogeneity evolution and fatigue life, the fatigue life of a superalloy and a pure copper under constant amplitude of tension-compression are studied by using the model of polycrystalline aggregate as the RVE with crystal plasticity [23,24]. Their investigation results show that the material deformation inhomogeneity at grain scale grows monotonically under cyclic loading. With the increase in the number of cycles, the strain inhomogeneity becomes more serious, and presents a correlation between the inhomogeneity and fatigue life. When the statistical standard deviation of longitudinal strain of the RVE, which reflected the inhomogeneity, is taken as the fatigue indicator parameter (FIP), it has a critical value to the fatigue failure occurrence. If the critical value is used to estimate the fatigue failure of the material, the result is approximately close to the test. In literature [23,24], the method is also used to predict the tension–compression fatigue life curves of a superalloy GH4169 and a steel HRB400, and the estimated errors between the prediction and test are found all to be within the acceptable range. However, the materials used in the above studies are all displaying approximate Masing behavior and, moreover, the shear strain fatigue is not researched.

In this paper, associating test and simulation, the study was carried on for a method to predict the low-cycle torsional fatigue of a steel with obvious non-Masing behavior. Shear fatigue at constant amplitude was investigated through simulation by a polycrystalline RVE as the material model, associated with crystal plasticity calculation referring tests. In view of the different elastic ranges and strain amplitudes, and the hysteresis behavior of non-Masing material, the material parameters of the crystal plasticity model were estimated from cyclic experimental hysteresis loops at different strain amplitudes. On the basis of cyclic deformation simulation tracking various strain fatigue experiments, the work was conducted to seek an improved FIP to measure the inhomogeneous deformation over the RVE. The critical value of FIP was determined according to the test at a single specified strain amplitude, and the shear fatigue life curve of the material was predicted and verified. Figure 1 shows the schematic diagram of the research model.

## 2. Experimental Programme

### 2.1. Materials and Specimen

The material used in this study was 30CrMnSiNi2A steel(Produced by Baoshan Iron & Steel Co., Ltd, Shanghai, China), supplied in the form of an annealed round bar of 20 mm diameter and the chemical composition (in wt %) is shown in Table 1. The heat treatment process was: heated to 890 °C in a box resistance furnace and oil quenched, then tempered of 400 °C, cooled in the air, finally. Figure 2 shows the microstructure of the material after heat treatment, which clearly reveals a tempered martensitic structure with distinct prior austenite grain boundaries marked partly with black lines shown in the zoomed view (at the left corner of Figure 2).

The typical dimension of the thin-walled tubular specimens is given inFigure 3. The inner and outer surfaces of the specimens were polished. We ensured that the collet of the testing machine could be clamped, and both ends of each thin-walled tubular specimen were blocked with metal plugs.

### 2.2. Torsional Fatigue Experiment

All the torsional fatigue experiments were carried out at room temperature on a servo-hydraulic tension-torsional material testing system (MTS809, TS Systems Corporation, Eden Prairie, MN, USA). The torsional angle of the gauge distance (25 mm) of the specimen was measured using an axial-torsional extensometer (MTS632.80F-04, MTS Systems Corporation, Eden Prairie, MN, USA). Based on the von Mises criterion used in this study, the equivalent strain is defined as $\varepsilon_{}^{eq}=\gamma /\sqrt{3}$ and the equivalent stress is $\sigma_{}^{eq}=\sqrt{3}\tau$. Therefore, the amplitude of equivalent shear strain $\varepsilon_{a}^{eq}$ and stress $\sigma_{a}^{eq}$ are the $\gamma_{a}/\sqrt{3}$ and $\sqrt{3}\tau_{a}$, respectively. The experiments were conducted under strain control and the equivalent shear strain amplitudes $\varepsilon_{a}^{eq}$ were as follows, respectively: 0.9%, 0.8%, 0.7%, 0.6%, 0.5% and 0.45% (corresponding shear strain amplitudes $\gamma_{a}$: 1.56%, 1.39%, 1.21%, 1.04%, 0.87% and 0.78%, respectively). The strain ratio $R_{\varepsilon }$ is −1. The experiment of each strain amplitude was repeated once, and the cyclic loading was performed with a sine wave. The torsional fatigue results are shown in Table 2. The listed test values of two specimens are separated by the slash symbol in the table.

### 2.3. Cyclic Softening and Non-Masing Behavior

Figure 4 shows the variation of maximum/minimum shear stress versus cycle numbers at different strain amplitudes. As seen in this figure, the cyclic stress response of the material has the obvious characteristic of cyclic softening in the whole process.

The Masing behavior of the material can be distinguished by the stable stress–strain hysteresis loops [25,26,27,28,29,30]. The lower tips of all the stable hysteresis loops from different strain amplitudes are transferred to a common origin. If the upper branches of the stable hysteresis loops overlap and emerge with a common curve, then the material can be considered to follow the Masing behavior. That is, the material elastic range of the hysteresis curve is independent of the strain amplitude or the yield stress of the material does not change with different strain amplitudes during the cycle. Otherwise, it can be said that the material shows non-Masing behavior whose hysteresis behavior is related to the amplitude of loading.

All the half-life hysteresis loops of the torsional cycles for different shear strain amplitudes of 30CrMnSiNi2A are transferred to a common origin (0, 0), as shown in Figure 5a and the ascending branches of the curves do not overlap obviously. The curves are obtained by translating each half-life hysteresis loop along the linear elastic portions so that the upper branches of all different shear strain amplitudes match, as shown in Figure 5b. The elastic range or yield stress varies greatly with different strain amplitudes, showing in Figure 5. That is, 30CrMnSiNi2A exhibits non-Masing behavior obviously.

## 3. Crystal Plastic Constitutive Model and Material Model

### 3.1. Crystal Plastic Constitutive Model with the Bauschinger Effect

Polycrystalline metal materials are composed of many grains with different sizes, shapes and initial orientations. The deformation of the material at the grain scale is characterized by crystal slip. Considering the micro-scale slip-deformation mechanism of the material slip system in the global coordinate system, the Euler’s velocity gradient tensor, which describes the crystal deformation, can be decomposed into elastic and plastic parts by using the multiplication of the deformation gradient tensor.
(1)$$\left\{\begin{array}{l} L=\dot{F}\cdot F^{-1}=L^{*}+L^{p},L^{*}={\dot{F}}^{*}\cdot F^{*}{}^{-1}\\ L^{p}=F^{*}\cdot {\dot{F}}^{p}\cdot {\dot{F}}^{p-1}\cdot F^{*}{}^{-1} \end{array}\right.$$
where $L^{*}$ and $L^{p}$ denote, respectively, the elastic part and the plastic part of Euler’s velocity gradient tensor L; $F^{*}$ and $F^{p}$, respectively, are the elastic part and the plastic part with respect to the deformation gradient tensor $F^{}$; $F^{*}$ reflects the rigid body rotation and the lattice elastic distortion during crystal deformation, $F^{p}$ describes the dislocation slip in crystal deformation.

For metallic materials, the elastic deformation is small, and the rate constitutive relationship can be expressed as: (2)$${\dot{\sigma }}^{J}=\overset{<4>}{C}:{\dot{\varepsilon }}^{e}=\overset{<4>}{C}:(\dot{\varepsilon }-{\dot{\varepsilon }}^{p})$$
where, ${\dot{\sigma }}^{J}$ is Jaumann stress rate, ${\dot{\varepsilon }}^{}$, ${\dot{\varepsilon }}^{e}$ and ${\dot{\varepsilon }}^{p}$ are total strain rate, elastic strain rate and plastic strain rate tensors, respectively; and $\overset{<4>}{C}$ is the fourth-order elastic constitutive tensor.

For the α-slip system, the Schmid tensor can be written as [31,32]:(3)$$P^{(\alpha )*}=\frac{1}{2}(m^{(\alpha )*}n^{(\alpha )*}+n^{(\alpha )*}m^{(\alpha )*})$$
(4)$$m^{(\alpha )*}=F^{*}\cdot m^{\left(\alpha \right)},\;\;n^{(\alpha )*}=n^{\left(\alpha \right)}\cdot F^{*-1}$$
where $n^{\left(\alpha \right)}$ denotes the unit vector that is orthogonal to the slip plane of the α-slip system; $m^{\left(\alpha \right)}$ is the unit vectors of the slip direction of the α-slip system. During the calculation, the Schmid tensor rotates with the crystal lattice. The increment of logarithmic plastic strain can be expressed as [33]:(5)$$\Delta \varepsilon^{p}=\sum_{\alpha =1}^{n}P^{(\alpha )*}\Delta \gamma^{\left(\alpha \right)}$$
where $\Delta \gamma^{\left(\alpha \right)}$ is got by integral of ${\dot{\gamma }}^{\left(\alpha \right)}$. The relationship between the Cauchy stress and resolved shear stress on α-slip system can be written as follows:(6)$$\tau^{\alpha }=P^{(\alpha )*}:\sigma$$

For the resolved shear strain rate of slip system, the back-stress $x^{\left(\alpha \right)}$ is introduced into the equation suggested by Hutchinson [34] to consider the Bauschinger effect [35]:(7)$${\dot{\gamma }}^{\left(\alpha \right)}={\dot{\gamma }}_{0}\mathrm{sgn}(\tau^{\left(\alpha \right)}-x^{\left(\alpha \right)}){\left|\frac{\tau^{\left(\alpha \right)}-x^{\left(\alpha \right)}}{g^{\left(\alpha \right)}}\right|}^{k}$$
where ${\dot{\gamma }}^{\left(\alpha \right)}$ and $\tau^{\left(\alpha \right)}$ are the resolved shear strain rate and resolved shear stress on α-slip system, respectively; ${\dot{\gamma }}_{0}$ denotes the reference shear rate, taken as the material constant to be determined; k denotes the rate correlation parameter of a material; $g^{\left(\alpha \right)}$ is the scalar function describing elastic range for α-slip system and its evolution is as follows using the equation suggested by Pan and Rice [36]:(8)$${\dot{g}}^{\left(\alpha \right)}(\gamma )=\sum_{\beta }^{\alpha }h_{\alpha \beta }(\gamma )\left|{\dot{\gamma }}^{\left(\beta \right)}\right|,\;\;\gamma =\int \sum_{\beta }^{\alpha }\left|d\gamma^{\left(\beta \right)}\right|$$
where $h_{\alpha \beta }^{}$ is the slip hardening moduli, which is described using the following expression proposed by Hutchinson [37]:(9)$$h_{\alpha \beta }(\gamma )=h(\gamma )[q+(1-q)\delta_{\alpha \beta }]$$
where q is a constant, 1≤q≤1.4. Chang and Asaro proposed that [38]:(10)$$h(\gamma )=h_{0}{\mathrm{sech}}^{2}(\frac{h_{0}\gamma }{\tau_{s}-\tau_{0}})$$
where $\tau_{0}$ is the critical resolved shear stress and $\tau_{s}$ is the saturation value; $h_{0}$ is the initial hardening rate. All these parameters are regarded as material constants.

The evolution of back-stresses, $x^{\left(\alpha \right)}$, is described by the expression [39,40]:(11)$${\dot{x}}^{\left(\alpha \right)}=a{\dot{\gamma }}^{\left(\alpha \right)}-c[1-e_{1}(1-\exp(-e_{2}\gamma ))]x^{\left(\alpha \right)}\left|{\dot{\gamma }}^{\left(\alpha \right)}\right|-px^{\left(\alpha \right)}$$
where a is material constant describing the linear hardening and c,p are the material constants for non-linear hardening of slip systems, $e_{1},e_{2}$ are the material constants representing the law of non-linear hardening saturation.

The calculation process of the above model is implemented by the user material subroutine (UMAT) of ABAQUS (Dassault Systems, Paris, France), which is detailed in [39,40]. However, it should be noted that the above crystal plasticity model is applicable to the materials showing Masing behavior. For non-Masing materials whose hysteretic behavior is related to strain amplitude, the model parameters are related to strain amplitude rather than constant. For example, the elastic range of the stable hysteretic curves for some non-Masing materials varies with the strain amplitude and $\tau_{s}$ is related to the strain amplitude. The hardening behavior of cycle with different strain amplitudes is obviously different. The correlation between the model parameters (a, c, $e_{1}$, $e_{2}$, p) describing the hardening behavior and the strain amplitude can be determined according to the actual test results.

### 3.2. Material Models

Taking assumption that the material is macroscopic initial isotropic and ignore the residual crystallographic texture, the polycrystalline RVE constructed by Voronoi aggregation is adopted as the material model, as shown in Figure 1. It contains 8000 elements of 8-node hexahedron and 9621 nodes which constructing 216 grains of the same anisotropic behavior. The orientation of each grain is generated randomly. And the microvoids and microcracks in the grains are ignored. According to continuum theory, the macroscopic Cauchy tensor Σ and logarithmic strain Ε are defined in terms of the load acting on the RVE and the resulting displacement, respectively.

The periodic boundary conditions are applied to the RVE models to satisfy the continuity of the material, ensuring that the RVE has the same deformation mode of the material where there is no separation or overlap. The periodic boundary conditions assume that any two parallel boundaries are composed of positive and negative parts ($\Omega_{i}^{+}\cup \Omega_{i}^{-}$), where each point $u_{i}^{+}\in \Omega_{i}^{+}$ has a counterpart point $u_{i}^{-}\in \Omega_{i}^{-}$ and their unit normal vectors must be satisfied $n_{i}^{+}=-n_{i}^{-}$, as shown in Figure 6a, and therefore the following unified periodic boundary conditions is obtained [41]:(12)$$u_{i}^{+}(x_{1},x_{2},x_{3})-u_{i}^{-}(x_{1},x_{2},x_{3})=c_{i}^{j}\left(i,j=1,2,3\right)$$
where $u_{i}^{+}(x_{1},x_{2},x_{3})$ and $u_{i}^{-}(x_{1},x_{2},x_{3})$ are the positive and negative displacements along the 1, 2 and 3 axes respectively, $c_{i}^{j}$ is constant. The periodic boundary conditions can be satisfied by applying linear equation constraints on the corresponding nodes on the surface of the RVE. In order to eliminate the rigid body displacement, the constraints are imposed on the three reference points A, C and D on RVE in Figure 6a, where $U_{1}^{A}=U_{2}^{A}=U_{3}^{A}=0$, $U_{1}^{D}=U_{3}^{D}=0$, $U_{1}^{C}=U_{2}^{C}=0$, respectively.

Applying periodic boundary conditions aforementioned can simulate the deformation of the macroscopic shear strain $\Gamma_{12}$ of material, exerting the vertical displacement $U_{2}$ at point B of the RVE, as shown in Figure 6b.

### 3.3. Material Parameters of Crystal Plasticity Model

For convenience of calculation in the finite element model (FEM), elastic constitutive tensor $\overset{<4>}{C}$ is usually expressed by matrix. For cubic crystal structure materials, there are only 3 independent stiffness coefficients in the matrix, namely, $C_{11}$, $C_{12}$ and $C_{44}$. Excepting these three elastic constants, the other material parameters of crystal plastic constitutive model need to be identified. The parameters were set as: ${\dot{\gamma }}_{0}=0.001$, k=200 (this means the material is approximately rate-independent); $e_{1}=e_{2}=0$, as the stable hysteresis behavior were considered; p=0, as creep and relaxation were ignored. The other parameters were calibrated by trial-error comparing cyclic experimental hysteresis loops, referring to the literature [39,40]. The model was originally proposed in literature [40], where the algorithm of the model was detailed and the physical meaning of each parameter was explained, and the method and process of calibrating the parameters were also introduced. Therefore, the details will not be elaborated on here.

The hysteresis behavior of 30CrMnSiNi2A was strain amplitude dependent, due to its non-Masing behavior. Parameters identification for RVE requires the reference of test hysteresis loops at different strain amplitudes, and the results obtained are shown in Table 3. Figure 7 shows the comparison between the experimental cyclic stress-strain curves of half-life with the hysteresis loops simulated by the RVE of the parameters in Table 3 and they are consistent.

According to the above analysis, considering correlation between the model parameters and strain amplitude, it was known that the hysteresis behavior of the non-Masing material can be described reasonably by numerical simulation using the RVE of crystal plasticity.

### 3.4. Validation of the Representative Volume Element (RVE)

The RVE be constructed in a random manner and as a material model it should have no significant difference in simulated behavior and deformation distribution. Its response to applied load should be close to the real material and not sensitive to the model varying and has no significant difference in statistics. In order to confirm this, a further validation is given in the following.

Figure 8a shows 5 RVEs with the same number of 8000 elements but containing 27, 64, 100, 216 and 512 grains, respectively. Figure 8b and c show their respective hysteresis loops and standard deviation curves at a shear strain amplitude $\gamma_{a}/\sqrt{3}=0.9\%$. As shown in the figures, the differences of hysteresis loop are marginal, and the curves are nearly the same except for the model containing 27 grains. The standard deviation curves are the same.

Figure 9a displays 3 models consisting of 1000, 8000 and 27,000 elements respectively, containing the same number of 216 grains. Figure 9b shows the comparison of their respective hysteresis behavior of strain amplitude $\gamma_{a}/\sqrt{3}=0.9\%$ and there is little difference in macroscopic hysteresis loops. The difference of standard deviation of the shear strain is also not considerable, as shown in Figure 9c.

To sum up, the model mentioned above was not sensitive to the random model structure varying and its behavior has no significant difference in statistics. It is necessary to point that the more grains and elements the model contains, the smaller the deviation of the calculation results. However, the larger the dimension of the model, the greater the amount of calculation.

## 4. Prediction of Low Cycle Fatigue Life Based on the Material Deformation Inhomogeneity

### 4.1. Analysis of Deformation Inhomogeneity Evolution under Shear Fatigue

The shear stress and strain distribution in the RVE varies with the cycle number. Figure 10 shows the contours at the peak shear stress and strain of cycles 7 and 560, corresponding to the cyclic strain amplitude of $\gamma_{a}/\sqrt{3}=0.9\%$. From the figure and the data in the legend box, the inhomogeneity of the shear strain distribution in the material increases obviously with the cycle, while it is very small for the shear stress distribution.

### 4.2. Statistical Analysis of Deformation Inhomogeneity Evolution

Under cyclic loading, there were significant differences in the distribution of meso-deformation and stress within the grain and neighboring grains, and statistical analysis was required to measure the variation of their distribution differences with cyclic number [41]. Statistic analysis for the Cauchy stress tensor $\sigma_{ij}$ and logarithmic tensor $\varepsilon_{ij}$ over RVE are carried out as follows:(13)$${\bar{\sigma }}_{ij}=\sum_{k=1}^{n_{RVE}}{\left(\sigma_{ij}\right)}_{k}p_{k}{\bar{\varepsilon }}_{ij}=\sum_{k=1}^{n_{RVE}}{\left(\varepsilon_{ij}\right)}_{k}p_{k}\;{\hat{\sigma }}_{ij}=\sqrt{\sum_{k=1}^{n_{RVE}}{\left(\sigma_{ij}\right)}^{2}{}_{k}p_{k}-{\bar{\sigma }}_{ij}},\;{\hat{\varepsilon }}_{ij}=\sqrt{\sum_{k=1}^{n_{RVE}}{\left(\varepsilon_{ij}\right)}^{2}{}_{k}p_{k}-{\bar{\varepsilon }}_{ij}}$$
where $n_{RVE}$ is the total number of finite elements in the RVE, ${\hat{\sigma }}_{ij}$ and ${\hat{\varepsilon }}_{ij}$ are the corresponding standard deviations, respectively; ${\bar{\sigma }}_{ij}$ and ${\bar{\varepsilon }}_{ij}$ are the mean values for $\sigma_{ij}$ and $\varepsilon_{ij}$, respectively; $p_{k}=\Delta V_{k}/V$ denoted the *k-th* volume as a percentage of total volume. The macroscopic mechanical behavior of materials can be described by the statistical mean values, and statistical standard deviation depict the difference of the distribution for the stress or strain at the grain-level. The higher the standard deviation is, the greater the dispersion within the RVE, and the deformation inhomogeneous of materials becomes more serious.

As we saw from the histogram as in Figure 11, the statistical distribution of shear stress and shear strain of each element in RVE is approximately Gaussian law. Moreover, the mean values of shear stress and shear strain at different cycle were almost the same; the standard deviation of shear stress varied little, but that of the shear strain varied by more than 10 times. This implies that under the macroscopic uniform shear cycle, the material deformation at the meso-scale (grain-scale) becomes more and more heterogeneity with the cycle, which reflects that the meso-structure of the material was changing constantly. It needs to be said that since the distribution and variation of stress depend on those of elastic strain (Hooke’s law), and the value and variation of material elastic strain are much less than that of plastic strain, it leads to very much smaller inhomogeneity for the shear stress than strain.

Accordingly, based on the above analysis and reference to the research in [39], the standard deviation of shear strain is used as the basis of the FIP to describe the shear fatigue damage in the following analysis.

### 4.3. Fatigue Indicator Parameter (FIP) Based on Evolution of Material Deformation Inhomogeneity

When the numerical simulation of pure copper under controlled cyclic symmetrical strain was carried out [39], the statistical standard deviation of the longitudinal strain was adopted as an indicator parameter to study the relationship between it and fatigue life. For low cycle fatigue of metal, the results show that the statistical standard deviation of the longitudinal strain has an approximate critical value and when the critical value is reached, the material will be damaged. It is considered that the fatigue life curve of materials can be reasonably predicted by using appropriate fatigue statistical parameters in the literature [39], however, the universality of this methodology needs to be further verified.

For the fatigue indicator parameter, there are two other aspects in the following to be considered: (1) if accumulative fatigue damage of the different cyclic is the same, whether the material rupture will occur immediately, the cyclic peak stress will play an important role; (2) for non-Masing materials, the yield stress is not the same at different cyclic strain amplitudes, which also has effects on the accumulative fatigue damage. Therefore, a new FIP which takes the standard deviation of the shear strain-${\hat{\gamma }}_{12}$ as the basis is proposed in this paper and is defined as:(14)$${\hat{\gamma }}_{12}^{I}=\frac{\Sigma_{\max}}{\Sigma_{0.1}}\cdot {\hat{\gamma }}_{12}$$
where $\Sigma_{\max}$ and $\Sigma_{0.1}$ are respectively the peak shear stress and the yield shear stress on half-life hysteresis loop (the definitions are shown in Figure 12a).

There is obviously cycle softening behavior of 30CrMnSiNi2a during the cycle. Figure 12b displays the hysteresis loops of different number of cycles at $\gamma_{a}/\sqrt{3}=0.8\%$ and the peak shear stress and yield shear stress of the hysteresis loop decrease gradually with the cycle. The hysteresis loop becomes stable with the increased number of cycles. The variation of $\Sigma_{\max}/\Sigma_{0.1}$ is shown in Table 4. Cyclic variation of other amplitudes is similarity. For simplify, the value of $\Sigma_{\max}/\Sigma_{0.1}$ at N/N*_f_* = 0.5 is uniformly substituted into Equation (13) for the calculation of ${\hat{\gamma }}_{12}^{I}$ in the paper.

### 4.4. Prediction and Validation of Shear Fatigue Life Curve

Figure 13a,b show the evolution curve of the parameters ${\hat{\gamma }}_{12}^{}$ and ${\hat{\gamma }}_{12}^{I}$ with shear cycle numbers; both of them increase monotonously with the increasing number of cycles. The logarithmic scale coordinates of the horizontal axis are utilized to distinguish better the cycle curves of different strain amplitudes.

In Figure 13a,b, the corresponding critical point can be determined on the parameter curve of a certain specified strain amplitude according to the test fatigue life, then draw a horizontal line across the critical point (the dotted line in the graph). The abscissa, $N_{fi}$, of the intersection point between the horizontal line and the curve corresponding to the *i*-*th* strain amplitude ${Ε}_{ai}$ is the predicted fatigue life of the corresponding shear strain amplitude and then the predicted life curve can be obtained (${Ε}_{ai}\sim N_{fi}$). According to this method, the fatigue failure is predicted by using the curves in Figure 13 and the critical value range of fatigue parameters ${\hat{\gamma }}_{12}^{}$ is 0.05649–0.12604, and -${\hat{\gamma }}_{12}^{I}$ is 0.08988–0.14204.

According to the maximum value (0.12604), median value (0.09126) and minimum value (0.05649) of ${\hat{\gamma }}_{12}^{}$ and the corresponding horizontal lines in Figure 13a, three of point sequences ($\gamma_{a}\sim N_{f}$) between strain amplitude and predicting fatigue life are obtained, and they are drawn in Figure 14a. Similarly, another three of point sequences can also be obtained according to the maximum value (0.14204), median value (0.11596) and minimum value (0.08988) of ${\hat{\gamma }}_{12}^{I}$ (cf. Figure 13b), and are drawn in Figure 14b. Thus, by using the above method the upper bound, median and lower bound predictions of fatigue life can be obtained.

### 4.5. Verification of Shear Fatigue Life Prediction Error

In order to confirm the rationality and effectiveness of the above fatigue life prediction curves, it is necessary to check the errors between the life predictions (see Figure 14a,b) and tests. It should be pointed out that the predictions of Figure 14a,b contain the predictions with the largest deviation degree (upper and lower bounds). This means that all predictions are rational if the upper and lower bounds of fatigue life prediction meet the error requirements.

The comparison of errors is shown in Figure 15, in which the abscissa and ordinate are the test and predicted values of fatigue life, respectively; the thick solid line is the ideal prediction, that is, the predicted life is completely consistent with the test; and the dashed lines on both sides represent the factor-of-two boundaries. Figure 15a shows that the error of the prediction by ${\hat{\gamma }}_{12}^{}$ relative to the test is mostly less than 2 times and no more than 2.5 times at most. However, for ${\hat{\gamma }}_{12}^{I}$, all fatigue life predictions fall within the factor of two scatter bands in comparison figures (see Figure 15b). That is, it is more reasonable to predict the shear fatigue life with the new FIP (${\hat{\gamma }}_{12}^{I}$).

## 5. Conclusions

In this research, the constant amplitude torsional fatigue test of 30CrMnSiNi2A steel was carried out first and then the shear cycle process was numerically simulated using the RVE constructed by Voronoi polycrystalline aggregates; subsequently the relationship between non-uniform deformation and fatigue failure under shear cycle was investigated. The following conclusions are drawn from this research:30CrMnSiNi2A steel exhibits non-Masing behavior under torsional cycle, its hysteretic behavior is associated with the strain amplitude, and its elastic ranges of different strain amplitudes are different. When the crystal plasticity RVE model is used to numerically simulate the constant amplitude shear cycle process, the correlation between the parameters for RVE and strain amplitude must be taken into account.There is a correlation between the evolution of deformation inhomogeneity and the shear fatigue life of material in torsion cycle.Using the standard deviation of shear strain (${\hat{\gamma }}_{12}^{}$) as the FIP, most of the fatigue life predictions fall within the factor of two scatter bands in comparison figures, and the maximum error is not more than 2.5 times compared with the test.If using the weighted standard deviation of the shear strain (${\hat{\gamma }}_{12}^{I}$) as the FIP, in which the ratio of peak stress to yield stress is taken into account, all fatigue life predictions fall within the factor of two scatter bands in comparison figures.

## Figures and Tables

**Figure 1 materials-14-01846-f001:**
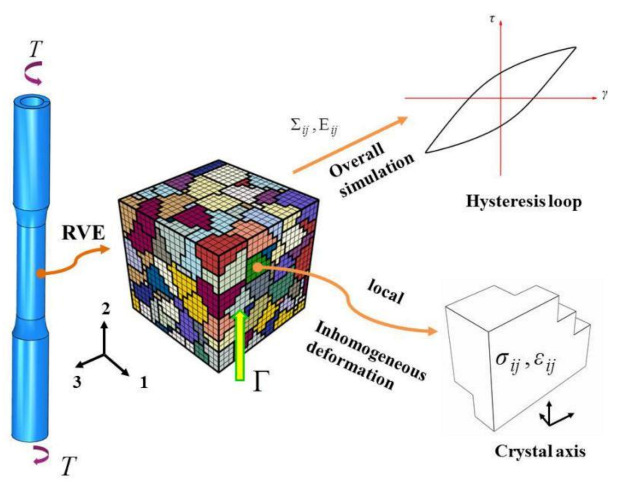
The schematic diagram of research the model.

**Figure 2 materials-14-01846-f002:**
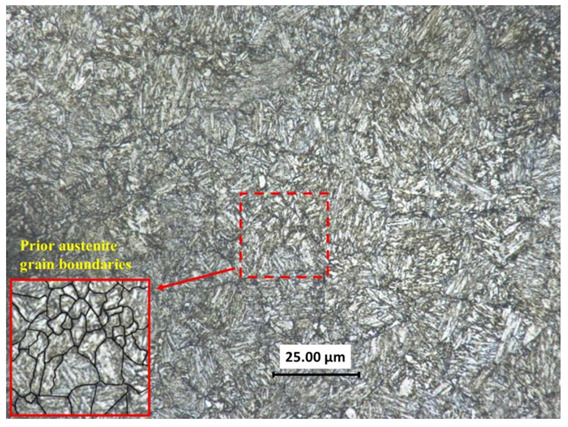
Microstructure of 30CrMnSiNi2a in as received condition.

**Figure 3 materials-14-01846-f003:**
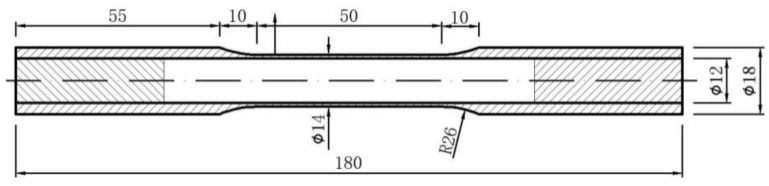
Geometries of the thin-walled tubular specimens used in the tests (mm).

**Figure 4 materials-14-01846-f004:**
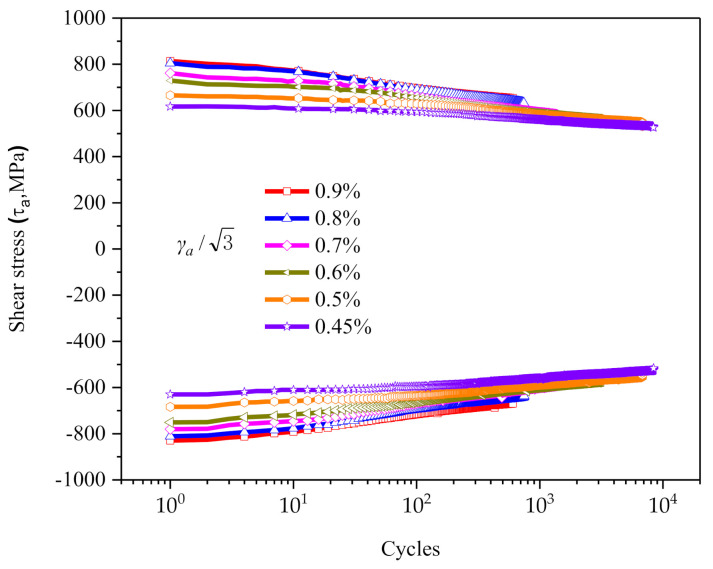
Variation of maximum/minimum shear stress versus number of cycles at different strain amplitudes.

**Figure 5 materials-14-01846-f005:**
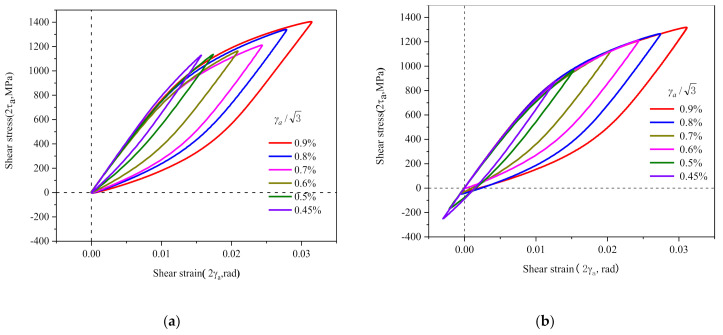
The half-life hysteresis loops for different shear strain amplitudes: (**a**) the lower tips of all hysteresis loops from different strain amplitudes are transferred to a common origin (0, 0); (**b**) translating along the linear elastic portions.

**Figure 6 materials-14-01846-f006:**
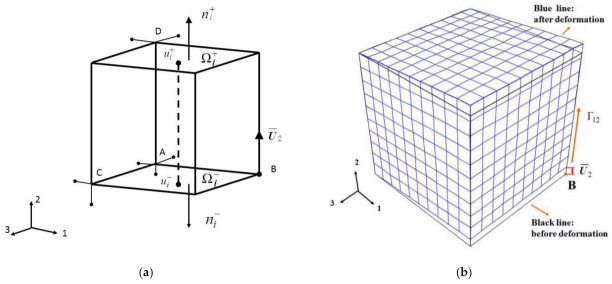
(**a**) Schematic diagram of representative volume element (RVE) boundary conditions; (**b**) RVE macroscopic deformation applied displacement $U_{2}$ at point B.

**Figure 7 materials-14-01846-f007:**
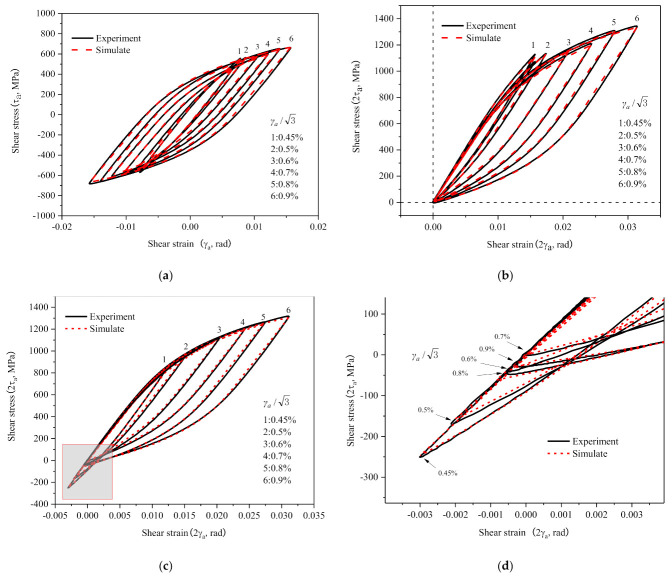
(**a**) The comparison of experiment and simulation of half-life hysteresis loops at different shear strain amplitudes; (**b**) the lowest point of the hysteresis loop overlaps at the origin (0, 0); (**c**) different shear amplitudes are translated along the elastic range; (**d**) locally enlarged view of block diagram in Figure 7c.

**Figure 8 materials-14-01846-f008:**
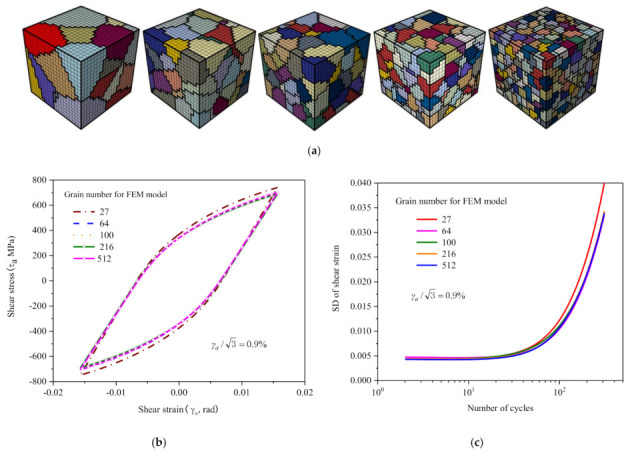
Models with the same number of 8000 elements: (**a**) containing 27, 64, 100, 216 and 512 grains; (**b**) comparison of hysteresis loops calculated by different models; (**c**) shear standard deviations vs. number of cycles for model varying number of grains.

**Figure 9 materials-14-01846-f009:**
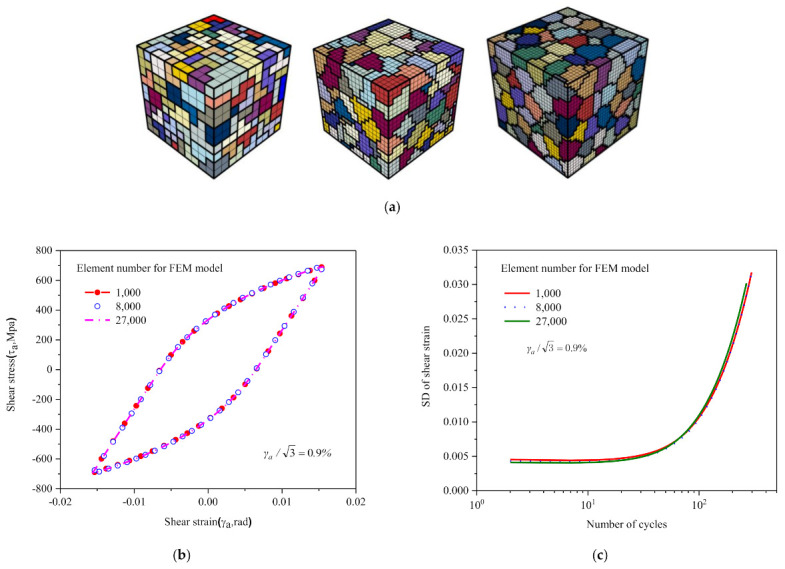
(**a**) The RVE model with the same number of 216 grains, containing 1000, 8000 and 27,000 elements respectively; (**b**) comparison of hysteretic curves calculated by different models; (**c**) comparison of shear standard deviations calculated by different models.

**Figure 10 materials-14-01846-f010:**
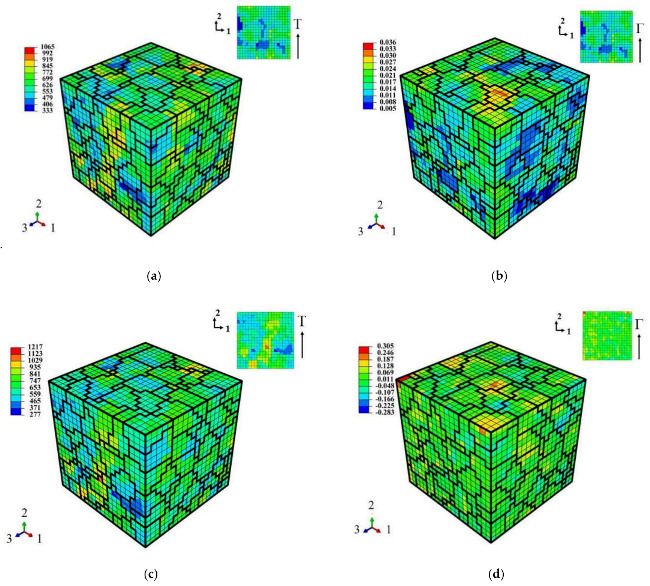
(**a**,**b**) Shear stress and strain contours at the shear stress peak of the 7th cycle, respectively; (**c**,**d**) Shear stress and strain contours at the shear stress peak of the 560th cycle, respectively. Shear strain amplitude is $\gamma_{a}/\sqrt{3}=0.9\%$.

**Figure 11 materials-14-01846-f011:**
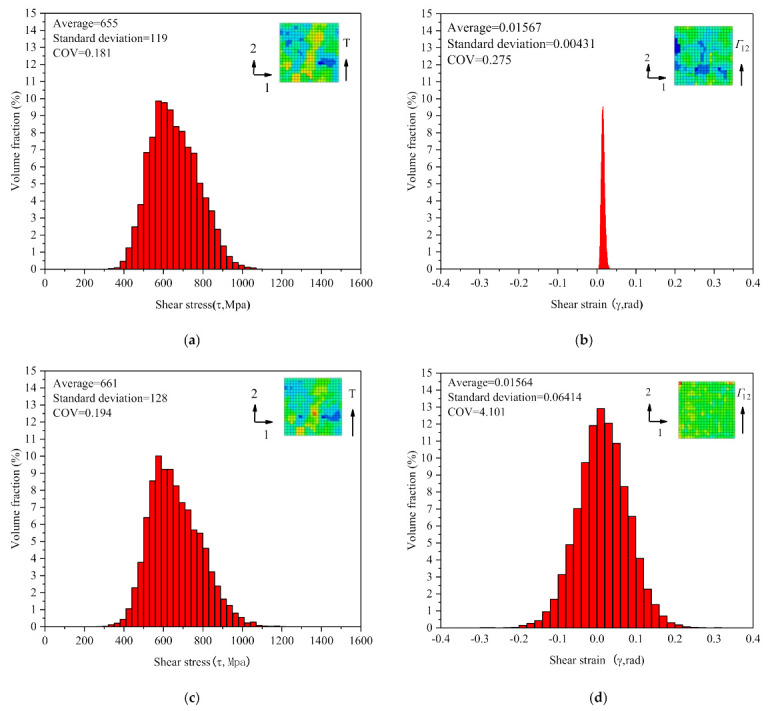
(**a**,**b**) Statistical distribution of shear stress and shear strain for the RVE at the shear stress peak of the 7th cycle, respectively; (**c**,**d**) statistical distribution of shear stress and shear strain for the RVE at the shear stress peak of the 560th cycle, respectively. Shear strain amplitude is $\gamma_{a}/\sqrt{3}=0.9\%$.

**Figure 12 materials-14-01846-f012:**
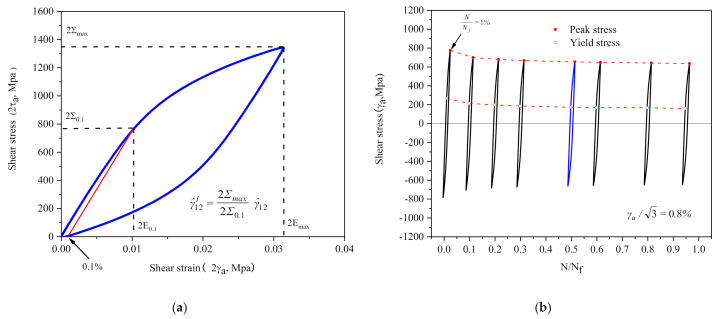
(**a**) Definition of shear yield stress; (**b**) the yield stress varies with the cycle numbers.

**Figure 13 materials-14-01846-f013:**
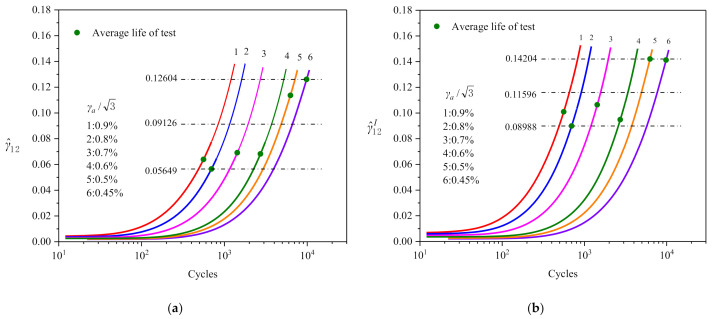
(**a**,**b**) Evolution of parameter ${\hat{\gamma }}_{12}^{}$ and ${\hat{\gamma }}_{12}^{I}$ with cycle numbers, respectively.

**Figure 14 materials-14-01846-f014:**
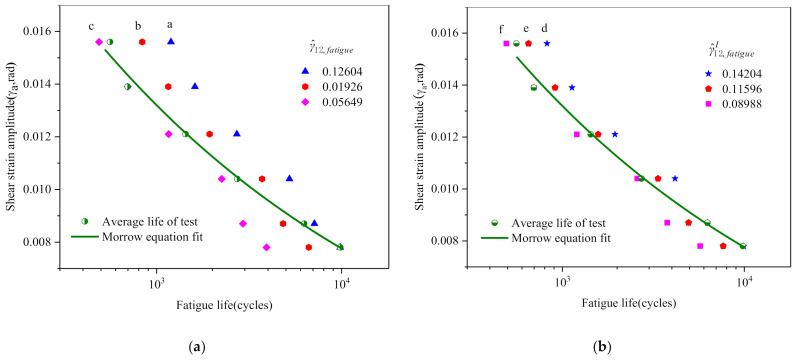
(**a**,**b**) Predicted life family lines by the maximum, median and minimum values of ${\hat{\gamma }}_{12}^{}$ and ${\hat{\gamma }}_{12}^{I}$, respectively.

**Figure 15 materials-14-01846-f015:**
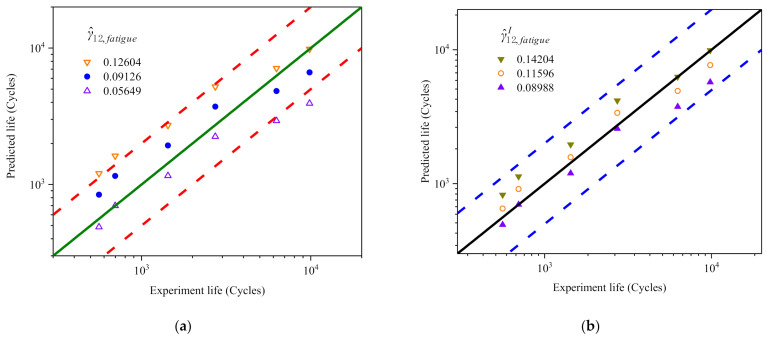
(**a**,**b**) Comparison between prediction and test of the parameters ${\hat{\gamma }}_{12}^{}$ and ${\hat{\gamma }}_{12}^{I}$, respectively. The maximum, intermediate and minimum values of their critical values are used as the criterion of fatigue damage. Strain amplitude $\gamma_{a}/\sqrt{3}$: 0.9%, 0.8%, 0.7%, 0.6%, 0.5% and 0.45%.

**Table 1 materials-14-01846-t001:** Chemical composition of 30CrMnSiNi2A (wt %).

C	Si	Mn	P	S	Ni	Cr	W	Cu	V	Ti
0.31	1	1.09	0.008	0.002	1.55	1.07	0.001	0.04	0.01	0.0058

**Table 2 materials-14-01846-t002:** The test torsional fatigue life of 30CrMnSiNi2A with $\gamma_{a}/\sqrt{3}$.

$\gamma_{a}/\sqrt{3}$	0.45%	0.5%	0.6%	0.7%	0.8%	0.9%
$N_{f}$	8392/11280	6630/5920	2306/3095	1458/1405	647/728	590/530
Average	9836	6275	2726	1432	698	560

**Table 3 materials-14-01846-t003:** Elastic constants and crystal plasticity parameters of 30CrMnSiNi2A at different shear amplitude.

$\gamma_{a}/\sqrt{3}$	Elastic Constants			Material Parameters of the Crystal Viscoplastic Model										
	$C_{11}$	$C_{12}$	$C_{44}$	$\tau_{0}$	$\tau_{s}$	$h_{0}$	a	c	p	$e_{1}$	$e_{2}$	${\dot{\gamma }}_{0}$	q	k
	GPa	GPa	GPa	MPa	MPa	MPa	GPa	MPa	s^−1^			s^−1^		
0.9%	276	207	138	320	380	80	3.2	30	0	0	0	0.001	1	200
0.8%	276	207	138	320	345	80	4.6	100	0	0	0	0.001	1	200
0.7%	276	207	138	285	300	80	6.3	100	0	0	0	0.001	1	200
0.6%	276	207	138	270	280	80	9.8	90	0	0	0	0.001	1	200
0.5%	276	207	138	320	340	80	11.7	120	0	0	0	0.001	1	200
0.45%	294	220	147	310	320	80	16.6	210	0	0	0	0.001	1	200

**Table 4 materials-14-01846-t004:** The $\Sigma_{\max}/\Sigma_{0.1}$ value at different N/N*_f_*, $\gamma_{a}/\sqrt{3}=0.8\%$.

N/N*_f_*	0.01	0.1	0.2	0.3	0.5	0.6	0.8	0.95
$\Sigma_{\max}/\Sigma_{0.1}$	1.49	1.53	1.55	1.56	1.57	1.58	1.58	1.60

## Data Availability

The data presented in this research are available on request from the corresponding author.

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
