# Peer review of "Torsional Fatigue Life Prediction of 30CrMnSiNi2A Based on Meso-Inhomogeneous Deformation"

_materials, 2021, doi:10.3390/ma14081846_

Round 1

Author Response

Response to Reviewer 1 Comments

We are very grateful to reviewer for the comments. The questions raised by the reviewer are answered one by one in the attached PDF file, please find it.
Thanks!

Reviewer 2 Report

The approach is inetersting, but the interpretation of plastic deformation at meso-scale can be influenced by the crustallographich texture that are present in the material. Have the authors supposed the material is isotropic? In this case they have to clearly specify that but also to indicate that the significance of the paper has to confined under this hypothesis. This assumption could be acceptable but it implies a loss about the general application of the results because the most of steels are affected by anisotropy. Thus, the authors have to clarify this point before the paper could be accepted.

Author Response

Response to Reviewer 2 Comments

We are very grateful to reviewer for the comments. The comments of the reviewer and our replies are listed below.

Point 1:  The approach is inetersting, but the interpretation of plastic deformation at meso-scale can be influenced by the crustallographich texture that are present in the material. Have the authors supposed the material is isotropic? In this case they have to clearly specify that but also to indicate that the significance of the paper has to confined under this hypothesis. This assumption could be acceptable but it implies a loss about the general application of the results because the most of steels are affected by anisotropy. Thus, the authors have to clarify this point before the paper could be accepted.

Response 1:

The influence of the crystallographic texture already in the material is ignored in this paper. The polycrystalline representative volume element (RVE) as a material model is constructed by Voronoi aggregation. This model assuming that the material is initial macroscopic isotropy in this paper. That is, if there is enough number of grains, the mechanical behavior of RVE will present approximately isotropic behavior. We consider the comments of the reviewer as correct. The crystallographic texture is produced in most steels during forming, although it cannot be completely removed through various heat treatments which will result in residual anisotropy. Certainly, it is better if this influence is considered in the analysis. For this, we added a corresponding supplementary explanation in the section “3.2 Material Models” in the revised version. See page 7, line 213 ~ 214.

Reviewer 3 Report

This paper presents the torsional fatigue behavior of 30CrMnSiNi2A steel and contains some interesting results, but the following points should be reconsidered.

1) Page 9, Line 259. The authors insisted that the differences of hysteresis loop are marginal in Fig. 8. However, there seems to be a threshold between 27 and 64. Please explain the reason for this.

2) It is well known that fatigue behaior and microstructure are closely related. In contrast, the results shown in Fig. 8 suggest that the effect of microstructure on the torsional fatigue behavior can be neglected. Please explain the reason for this.

3) Page 11, Line 275. The authors described that the inhomogeneity of the shear strain distribution in the material increases obviously with the cycle, while it is very small for the shear stress distribution. What does this result suggest?

Author Response

Response to Reviewer 3 Comments

Firstly, we are very grateful to reviewer for the comments. The comments of the reviewer and our replies are listed below.

Point 1: 1) Page 9, Line 259. The authors insisted that the differences of hysteresis loop are marginal in Fig. 8. However, there seems to be a threshold between 27 and 64. Please explain the reason for this.

Response 1:

(1) For the representative volume element (RVE) of Voronoi polycrystalline aggregates, each grain of the RVE with randomly orientation behaviors in anisotropic. In the case of RVE containing a small number of grains, the model response will be significantly different for the differences of anisotropy between grains. However, with the number of grains increasing, the apparent response difference caused by the random orientation of grains will be reduced.

(2) There seems to be a threshold between 27 and 64 grains in the model, from the calculated result, but we can’t be sure. Due to the diversity of randomly generated models, we cannot give a definite conclusion to this currently. It may require further calculation and theoretical analysis, but it seems that cannot be solved nowadays in the discussed scope of this paper. Thanks to reviewer giving us this suggestion.

Point 2: 2) It is well known that fatigue behavior and microstructure are closely related. In contrast, the results shown in Fig. 8 suggest that the effect of microstructure on the torsional fatigue behavior can be neglected. Please explain the reason for this.

Response 2:

(1) The material mechanical behavior is closely related to its microstructure. The material microstructure is very complex, involving different levels and sizes. This paper is focus mainly on the structure of polycrystalline aggregate of materials, without considering the factors such as reinforcement phase, cracks, voids, inclusions and grain boundaries. All these factors have certainly impact on the behavior of materials, especially material anti-fatigue properties, making the materials be enhanced or weakened.

(2) The polycrystalline representative volume element (RVE) is taken as a material model that simulates each grain with an equivalent anisotropic single crystal. The study aim of the paper is to instigate whether the model has the ability to reflect the law of the fatigue life and damage evolution of materials under cyclic loading.

(3) The more grains of the model contain, the smaller the deviation of the calculation results of the model. The number of grains in the model does not mean the microstructure of the material is different, but only reflects the calculation dimension of the model. However, the larger the number of grains, the greater the amount of calculation. And if it is too large it will be not advisable to be applied to simulate the cycle process of fatigue. Therefore, it is necessary to look at what dimension of the model can reasonably reflect the mechanical behavior of the materials. Figure 8 indicate that if the model grain number is not less than 100, by which the deviation of the calculation result can be controlled within a small range.

Based on above, we added a corresponding supplementary explanation in the section “3.4 Validation of the RVE”. See page 11, line 282~ 285.

Point 3: 3) Page 11, Line 275. The authors described that the inhomogeneity of the shear strain distribution in the material increases obviously with the cycle, while it is very small for the shear stress distribution. What does this result suggest?

Response 3:

According to the constitutive relation of the material, the stress depends on the elastic strain (Hooke’s law). Therefore, in the representative volume element (RVE), the stress distribution and its variation depend on the elastic strain distribution and its variation of the model. However, in peak strain during low cycle fatigue test, the variation of material elastic strain is much less than that of plastic strain and the plastic strain is the main component. Therefore, the variation of shear stress distribution is much smaller than that of shear strain distribution. This result means that the material deformation inhomogeneity caused by fatigue cycle is greater than the stress inhomogeneity, that is, the fatigue damage of material can be measured by the material deformation inhomogeneity.

The authors added a corresponding supplementary explanation in the section “4.2 Statistical analysis of deformation inhomogeneity evolution”. See page 11, line 317~ 320.

Round 2

Reviewer 1 Report

The authors have answered all my questions comprehensively and introduced necessary corrections into the manuscript. In my opinion, the paper can be accepted in the present form.

Reviewer 2 Report

The explanation provided have satisfied me.